# Improved [^18^F]FDG PET/CT Diagnostic Accuracy for Infective Endocarditis Using Conventional Cardiac Gating or Combined Cardiac and Respiratory Motion Correction (CardioFreeze^TM^)

**DOI:** 10.3390/diagnostics13193146

**Published:** 2023-10-07

**Authors:** D. ten Hove, B. Sinha, J. H. van Snick, R. H. J. A. Slart, A. W. J. M. Glaudemans

**Affiliations:** 1Department of Nuclear Medicine and Molecular Imaging, University Medical Center Groningen, University of Groningen, 9700 RB Groningen, The Netherlands; j.h.van.snick@umcg.nl (J.H.v.S.); r.h.j.a.slart@umcg.nl (R.H.J.A.S.); a.w.j.m.glaudemans@umcg.nl (A.W.J.M.G.); 2Department of Medical Microbiology and Infection Prevention, University Medical Center Groningen, University of Groningen, 9700 RB Groningen, The Netherlands; b.sinha@umcg.nl; 3Biomedical Photonic Imaging Group, Faculty of Science and Technology, University of Twente, 7522 NH Enschede, The Netherlands

**Keywords:** infective endocarditis, [^18^F]FDG PET/CT, cardiac motion correction

## Abstract

Infective endocarditis (IE) is a serious and diagnostically challenging condition. [^18^F]FDG PET/CT is valuable for evaluating suspected IE, but it is susceptible to motion-related artefacts. This study investigated the potential benefits of cardiac motion correction for [^18^F]FDG PET/CT. In this prospective study, patients underwent [^18^F]FDG PET/CT for suspected IE, combined with a conventional cardiac gating sequence, a data-driven cardiac and respiratory gating sequence (CardioFreeze^TM^), or both. Scans were performed in adherence to EANM guidelines and assessors were blinded to patients’ clinical contexts. Final diagnosis of IE was established based on multidisciplinary consensus after a minimum of 4 months follow-up and surgical findings, whenever performed. Seven patients participated in the study, undergoing both an ungated [^18^F] FDG-PET/CT and a scan with either conventional cardiac gating, CardioFreeze^TM^, or both. Cardiac motion correction improved the interpretability of [^18^F]FDG PET/CT in four out of five patients with valvular IE lesions, regardless of the method of motion correction used, which was statistically significant by Wilcoxon’s signed rank test: *p* = 0.046. In one patient the motion-corrected sequence confirmed the diagnosis of endocarditis, which had been missed on non-gated PET. The performance of the two gating sequences was comparable. In conclusion, in this exploratory study, cardiac motion correction of [^18^F]FDG PET/CT improved the interpretability of [^18^F]FDG PET/CT. This may improve the sensitivity of PET/CT for suspected IE. Further larger comparative studies are necessary to confirm the additive value of these cardiac motion correction methods.

## 1. Introduction

Infective endocarditis (IE) is a life-threatening infection of the endothelial surface of the heart, most commonly the heart valves or implanted cardiac materials. The disease is relatively rare, with a yearly incidence estimated at 3–10 per 100,000 people [1]. However, with the increasing possibilities for cardiac interventions, the population at risk for IE has greatly increased over the past decades and the disease has become increasingly associated with healthcare exposure. IE presents clinicians with a serious diagnostic challenge. The clinical presentation is highly variable and establishing the diagnosis can be difficult even with advanced imaging modalities and microbiological diagnostics. This difficulty is exacerbated by the need for prolonged and intensive treatment to improve patient outcomes. Consequently, mortality rates for IE are high. In a European registry published in 2019, in-hospital mortality from IE ranged from 15% to almost 20% depending on the presence of prosthetic valves or cardiac-implanted electronic devices (CIEDs) such as pacemakers or implantable cardioverter-defibrillators [2]. One-year mortality rates are even higher, exceeding 30% [1].

18-Fluorine fluorodeoxyglucose positron emission tomography–computed tomography ([^18^F]FDG-PET/CT) is a valuable diagnostic tool for the evaluation of suspected IE, and use of this technique as part of the diagnostic work-up for IE is recommended by both the American Heart Association (AHA) and the European Society of Cardiology (ESC) [3,4]. Additionally, the ESC has incorporated findings indicative of IE on [^18^F]FDG-PET/CT as a major criterion for diagnosis [3,5]. However, it has certain limitations, one of which is that it is susceptible to cardiac-motion-related artifacts. This susceptibility might explain the low sensitivity of [^18^F]FDG-PET/CT IE for suspected native valve endocarditis (NVE). For this group, [^18^F]FDG-PET/CT achieves a sensitivity of 36% [6], compared to 86% for suspected prosthetic valve endocarditis (PVE) [7]. The explanation for this difference could be that for native valves, the main areas of vulnerability are the highly mobile valve leaflets. For prosthetic valves, on the other hand, the area most vulnerable for bacterial infection is the annulus of the implanted valve, which is much less mobile during the cardiac cycle and in general larger in size. This might also help explain the limited sensitivity of [^18^F]FDG-PET/CT for the diagnosis of endocarditis in patients with cardiac-implanted electronic devices (CIEDIE) when lead involvement is suspected [8], as the intracardiac leads may likewise be affected by cardiac motion.

Cardiac motion correction in [^18^F]FDG-PET/CT may not routinely be performed in clinical practice, but the option to do so is widely available in current digital PET/CT camera systems. Conventionally, this is performed through a single-gated cardiac motion correction sequence using an ECG trigger. This is effective, but this approach leads to a loss of signal-to-noise ratio, as not all counts are used to build the image. How much signal is lost depends on whether single or dual gating is used; with single gating, 12.5% of the signal typically remains, while for dual gating as little as 4.2% of the original count rate is available per gate [9]. Furthermore, respiratory blurring of the image is still present [9]. Siemens digital PET/CT systems offer the CardioFreeze^TM^ technique as an alternative. This motion correction sequence corrects for both cardiac and as well for respiratory motion using list-mode data in combination with a deblurring algorithm using mass-conservation-based optical flow [9]. This achieves both cardiac and respiratory motion correction while simultaneously using all available counts, leading to improved image quality [9]. The aim of this study was to evaluate the impact of conventional cardiac gating and CardioFreeze^TM^ for their impact on the interpretability of [^18^F]FDG PET/CT in suspected cases of IE.

## 2. Materials and Methods

### 2.1. Patients

This was a cross-sectional prospective exploratory study conducted in a convenience sample of 7 patients with a suspicion of IE according to the criteria published by the British Society for Antimicrobial Chemotherapy (BSAC) [10]. Patients were included from June 2020 to October 2022. General demographic data of the included patients were recorded. These included information about any cardiac-implanted materials, previous cardiac surgeries, presenting symptoms in the current episode, findings from clinical and laboratory investigations including inflammatory markers, blood cultures, as well as findings during surgery whenever performed and any findings from additional analysis of removed materials. Additionally, potential confounders were documented. These included the duration of IV antibiotic use prior to PET/CT, intervals between the most recent cardiac surgical procedure if performed, and the use of surgical adhesives during valve implantation, such as BioGlue^®^.

### 2.2. Ethical Approval

The study was approved as a non-WMO study by the UMCG Medical Ethics Review Board (protocol nr. M16.198401). All patients provided written informed consent to use the conventional gating sequence, CardioFreeze^TM^, or both while undergoing [^18^F]FDG-PET/CT, which was indicated as part of their routine diagnostic work-up for suspected IE.

### 2.3. [^18^F]FDG PET/CT Protocol

#### 2.3.1. Patient Preparation

The preparation, acquisition, and evaluation of [^18^F]FDG PET/CT were all performed in accordance with the European Association of Nuclear Medicine (EANM) guidelines [11]. 

A 24 h high-fat, low-carbohydrate diet and 6 h fasting time prior to the scan were used in all patients. Additionally, if there were no contra-indications against its use, an intravenous heparin injection of 50 IU/kg was given 15 min prior to the scan to further suppress physiological [^18^F]FDG uptake. All scans were acquired on the Biograph Vision digital PET/CT system (Siemens Healthineers, Knoxville, TN, USA). Gating was performed using co-registration of an ECG, both for regular gating (single gate) and CardioFreeze^TM^, and in which the sequences were performed in a single-bed position for 10 min, immediately after the regular non-gated [^18^F]FDG PET/CT scan had been completed.

#### 2.3.2. [^18^F]FDG PET/CT Visual Evaluation

The Syngo.via VB40 software (Siemens Healthineers, Knoxville, TN, USA) was used for the visual analysis. Two experienced nuclear medicine physicians (RS, AG) performed a consensus reading of the scans, blinded to the clinical context of the patients. The image quality of both non-gated and gated scans was rated as 0 (uninterpretable), 1 (poor), 2 (moderate), 3 (good), or 4 (excellent). The interpretation of [^18^F]FDG PET/CT was based on the pattern, intensity, and extent of FDG uptake in lesions adjacent to the suspected valve(s) and/or any other implanted materials, in accordance with guidelines from the EANM [11]. Both attenuation-corrected (AC) and uncorrected (NAC) images were used for the evaluation.

### 2.4. Final Diagnosis

The diagnosis of infective endocarditis was established by one of two methods. In cases where surgery had been performed, macroscopic signs of infection during surgery or confirmation of infection on subsequent histology, pathology, or cultures were considered the gold standard for diagnosis. In cases where surgery had not been performed, a consensus diagnosis was made by a multidisciplinary endocarditis team based on all available clinical information. This included findings from patient history, physical examination, blood cultures, and molecular diagnostics, and various imaging techniques such as transthoracic and transoesophageal echocardiography (TTE and TEE), diagnostic CT angiography, and [^18^F]FDG PET/CT. The final diagnosis was further based on patient outcome during a minimum of 4 months of follow-up, which altogether served as a composite gold standard for the diagnosis.

### 2.5. Statistics

Statistical analysis was performed using SPSS version 26.0 (IBM Corp, Armonk, NY, USA). Continuous variables were presented as mean ± standard deviation for normally distributed data or as median ± interquartile range (IQR) for non-normally distributed data. Categorical variables were reported as frequencies. The diagnostic performance of [^18^F]FDG PET/CT with and without cardiac motion correction was reported descriptively. Sensitivity and specificity were not reported in this series due to the limited sample size and the heterogeneity of the included cases (e.g., both suspected NVE and PVE). The interpretability scores of [^18^F]FDG PET/CT with and without cardiac motion correction were reported and compared using Wilcoxon’s signed rank test. The conventional and data-driven motion correction sequence were not compared to each other due to the limited number of cases in which both were performed.

## 3. Results

Seven patients underwent [^18^F]FDG PET/CT with cardiac motion correction using either a single-gated sequence, CardioFreeze^TM^, or both. In two patients there was no FDG uptake indicative of infective endocarditis, neither on the ungated PET/CT nor on the CardioFreeze^TM^ images. In these two patients, further diagnostic work-up found no abnormalities consistent with IE and the diagnosis was rejected in both cases. In the other five subjects, intracardiac uptake was present on both ungated and gated PET/CT, but the intensity and interpretability of the findings differed. An overview of the patient characteristics is shown in Table 1, while the details of their clinical presentation and the findings during their evaluation and follow-up are presented below.

Patient 1 was a 78-year-old male with a history of TAVI for aortic valve stenosis and DDDR pacemaker implantation due to post-surgical AV block 10 months prior to presentation for the current episode. The patient presented with anemia, right-sided heart failure, and concentration problems. The CT-cerebrum showed signs of non-recent ischemic damage in the left frontoparietal cortex. Blood cultures demonstrated a *Staphylococcus saprophyticus* bacteremia. An echocardiogram showed thickening of the TAVI valve with a large mobile structure (>30 mm), suspected to be either new thrombus or infective endocarditis. Conventional ungated PET with CTA confirmed infective endocarditis with intense uptake at the TAVI valve. Likewise, CardioFreeze^TM^ confirmed IE with intense uptake, while additionally showing that the area with uptake moved with the cardiac cycle (see Figure 1). Four days after [^18^F]FDG PET/CT, the vegetation could no longer be detected on echocardiography—likely due to luxation. However, the patient’s clinical condition improved under antibiotics. Due to high risk of severe complications from surgery and the patient’s fragile clinical condition, treatment was continued conservatively with lifelong antibiotic suppression. During follow-up, no recurrence of infectious complaints occurred. 

Patient 2 was a 76-year-old female who was admitted with acute-phase reaction and symptoms of right hip and groin pain, limited neck mobility, and a urinary tract infection. Blood cultures showed *Staphylococcus aureus* bacteremia, indicative of disseminated infection. The patient started with antibiotics and underwent a diagnostic work-up in accordance with the in-hospital endocarditis protocol. Transthoracic and transesophageal echocardiography showed no abnormalities consistent with IE and [^18^F]FDG PET/CT showed no signs of endocarditis, both on gated and non-gated images. The diagnosis of endocarditis was rejected and MRI showed spondylodiscitis with a prevertebral fluid collection and an abscess of the right iliacus muscle as an alternative explanation for the patient’s symptoms. The patient was successfully treated for complicated *S. aureus* bacteremia with appropriate intravenous antibiotics and was discharged with continued antibiotics at home. 

Patient 3 was a previously healthy 18-year-old male who presented with fever and fatigue. A physical examination revealed a systolic murmur, blood cultures showed *Staphylococcus aureus* bacteremia, and echocardiography revealed a vegetation on the tricuspid valve. No risk factors for infection were identified, except for chronic paronychia of an ingrown toenail as a possible point of entry. The patient had no history of congenital heart disease, no family history, and no risk behaviors. Non-gated [^18^F]FDG PET with low-dose CT confirmed the diagnosis of native valve endocarditis, showing intense FDG uptake at the tricuspid valve and pulmonary septic emboli. Cardiac gating markedly improved the scan interpretability and demonstrated fluttering of the vegetation at the tricuspid valve with intense FDG uptake, as shown in Figure 2. The patient was treated conservatively with appropriate intravenous antibiotics and made a full recovery.

Patient 4 was an 80-year-old male with a history of cardiac ischemia treated by CABG and valvular dysfunction for which he underwent valve replacement with a biological AVR and later a TAVI aortic valve replacement. Two years prior to the current episode, the patient was diagnosed with *Cutibacterium* (formerly: *Proprionibacterium*) *avidum* endocarditis, which was treated conservatively with lifelong suppression with amoxicillin. This time, the patient presented with acute heart failure without infectious symptoms. Blood cultures remained negative but echocardiography showed new, severe insufficiency of the aortic valve, and tricuspid valve insufficiency. [^18^F]FDG PET/CT was performed, showing equivocally increased FDG uptake around the valve, both on gated and non-gated PET/CT (Figure 3). Due to a lacking sufficient alternative explanation of his signs and symptoms, the patient was treated with appropriate antibiotics and he underwent elective AVR and TVR 6 weeks later. The aortic valve showed severe damage consistent with infection and 16S-rDNA PCR with sequencing detected *Cutibacterium avidum*, confirming the diagnosis of IE. The patient was discharged and made a full recovery.

Patient 5 was an 83-year-old female with a history of TAVI, who presented with septic arthritis caused by *Enterococcus faecalis*, which was also isolated from her blood cultures. Aortic valve endocarditis was suspected, and confirmed by TEE. FDG PET/CT was performed to evaluate the presence of disseminated disease. Mild FDG uptake was observed on non-gated [^18^F]FDG PET/CT at the aortic valve, insufficient for confirmation of endocarditis due to the absence of uptake on NAC images. CardioFreeze^TM^ revealed more pronounced uptake moving with the cardiac cycle, confirming the diagnosis (see Appendix A). The patient was treated conservatively with appropriate intravenous antibiotics and lifelong suppressive therapy.

Patient 6 was a 23-year-old female with a history of complex congenital heart defects, for which a Fontan circulation and a bidirectional Glenn shunt had been established. The patient presented with *Staphylococcus aureus* bacteremia. No signs of endocarditis were seen on either TEE or non-gated [^18^F]FDG PET/CT or CardioFreeze^TM^, but an immobile lesion with increased FDG uptake was noticed at the site of an epicardial surgical membrane from previous surgery. The image quality was not changed by application of motion correction and CardioFreeze^TM^ confirmed the immobility of the lesion throughout the cardiac cycle. The patient was successfully treated with appropriate intravenous antibiotics, followed by lifelong oral antibiotic suppressive therapy.

Patient 7 was a 22-year-old male with a history of corrected Fallot’s tetralogy and a Contegra^TM^ graft for his pulmonic valve. He presented with *Aggregatibacter aphrophilus* (HACEK group) bacteremia. No clear point of entry could be identified. Echocardiography showed no signs of endocarditis. [^18^F]FDG PET/CT showed increased uptake at the pulmonic valve, suspect for endocarditis. CardioFreeze^TM^ showed subtle fluttering of the lesion, increasing the suspicion that the uptake was caused by a vegetation and further confirming the diagnosis. The patient was treated with 6 weeks of appropriate intravenous antibiotics and he made an uneventful recovery with no further events during follow-up.

Overall, in this series, [^18^F]FDG PET/CT without cardiac motion correction identified three out of five cases in which IE was confirmed as the final diagnosis. When cardiac motion correction was applied, this established the diagnosis in one additional patient. Two patients in this series did not have IE as their final diagnosis and [^18^F]FDG PET/CT findings were negative in both cases, regardless of whether cardiac motion correction was applied. 

The mean interpretability scores were 3.0 without cardiac motion correction. When motion correction was applied, this increased the mean interpretability scores to 3.6, which was a statistically significant improvement by Wilcoxon’s signed rank test (*p* = 0.046) and judged as clinically useful by the nuclear imaging experts.

An overview of all patient findings, including those of the different PET/CT sequences, is shown in Table 2. Additionally, Cine loop videos are provided as Appendix A for all included patients.

## 4. Discussion

In this exploratory study, motion correction of [^18^F]FDG PET/CT using either conventional single-gate gating or CardioFreeze^TM^ improved the interpretability in four out of five cases with valvular IE lesions, confirming the diagnosis in one case where non-gated PET had missed the diagnosis. Both motion correction methods performed equally well. These findings are in line with an earlier study examining the role of cardiac motion correction [12] that likewise found indications that the technique leads to improved interpretability and diagnostic accuracy. All scans were performed in accordance with EANM recommendations and the [^18^F]FDG PET/CT evaluation was performed by two experienced nuclear physicians, blinded to the patients clinical context, which adds further robustness to our findings. 

Cardiac motion correction and visualization of cardiac motion across the different gates allows the clinician to avoid the blurring that cardiac motion ordinarily creates in ungated [^18^F]FDG PET/CT, while it also allows for tracking the cardiac movement over time. This can help differentiate between physiological uptake caused by, e.g., septal and/or ventricular myocardial activity and infectious processes, based on the movement patterns of areas with increased uptake. This leads to higher image quality scores given to the gated images compared to the ungated ones. This difference was statistically significant, and while this should be interpreted with appropriate caution given the small sample size and the heterogeneity of the sample, it is a promising finding. Additionally, as shown in various cases in this study, cardiac gating may allow for better direct visualization of valvular vegetations, a known difficulty in [^18^F]FDG PET/CT [6]. 

As noted in the results section, there were no overt differences in image quality for images obtained through CardioFreeze^TM^ compared to conventional cardiac gating in the three cases where both were obtained. The reason for this is not completely clear, but a potential explanation may be that the time taken for the image acquisition in the single-bed position was long enough that the difference in photon counts were no longer decisive for determining image quality. This might mean that shorter acquisition times might be possible when using this sequence. Furthermore, CardioFreeze^TM^ was used with ECG co-registration and, theoretically, this sequence could also work without co-registration, which may render it easier to apply in clinical practice. 

Furthermore, the patient burden for using these motion correction sequences is very limited, since only ECG co-registration and one additional low-dose CT is necessary. The radiation burden of the low-dose CT is negligible (~0.5 mSv) and the required time to perform the extra sequences is no longer than 10 min. In the future, even this negligible burden may disappear when it becomes possible to perform cardiac motion correction sequences simultaneously, e.g., with the regular scan in large field-of-view PET/CT camera systems. 

### Limitations

This study used a relatively small convenience sample, limiting the generalization of our findings. This also precluded further in-depth analysis of factors that might have affected the efficacy of cardiac motion correction strategies for improving image interpretability, such as the presence of prosthetic valves or other intracardiac implanted materials. This would be an interesting area for future studies. 

Due to the exploratory nature of the study, no power analyses were performed. An important consideration for future studies is that the prevalence of IE at the time of [^18^F]FDG-PET/CT is not known, due to differences in timing of the PET/CT and other data (e.g., NVE vs. PVE, pathogen, other diagnostic tests performed, prior antimicrobial treatment, etc.). These factors are, among others, determinants of the a priori chance of disease presence and of [^18^F]FDG-PET/CT’s diagnostic performance. Although we estimated a sample size of 58 per arm based on our preliminary data using Yates’ continuity correction in a paired study design (β = 80%, α = 0.05), this remains a rough estimate that would require interim analysis for a more accurate calculation [13].

Despite the limitations, our results indicate that cardiac motion correction techniques for [^18^F]FDG PET/CT show promise for use in clinical practice with a negligible patient burden.

## 5. Conclusions

In this exploratory study, cardiac gating and/or CardioFreeze^TM^ improved interpretability of [^18^F]FDG PET/CT and confirmed a diagnosis of IE that had been missed on ungated [^18^F]FDG PET/CT. Cardiac motion correction allowed for visualization of vegetation fluttering. CardioFreeze^TM^ performed equally compared to conventional cardiac gating. Larger prospective study cohorts are needed to validate our findings. A visual representation of our study findings can be found in the graphical abstract.

## Figures and Tables

**Figure 1 diagnostics-13-03146-f001:**
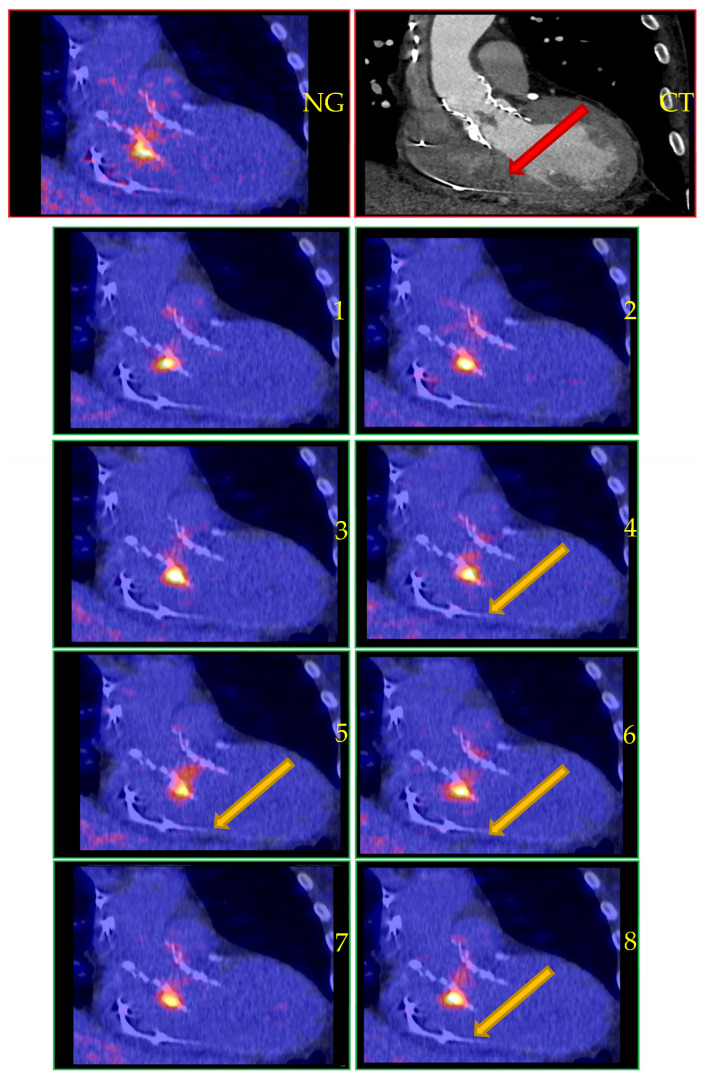
[^18^F]FDG PET/CT aortic valve (TAVI) endocarditis, as shown by conventional PET and CTA (top two images) and CardioFreeze^TM^ (panels 1–8). Legend: NG, non-gated PET/CT; CT, CTA. Note the mobile lesion with increased FDG uptake in panels 4–6 and 8 (arrows), corresponding with the vegetation also seen on CTA performed the same day (red arrow).

**Figure 2 diagnostics-13-03146-f002:**
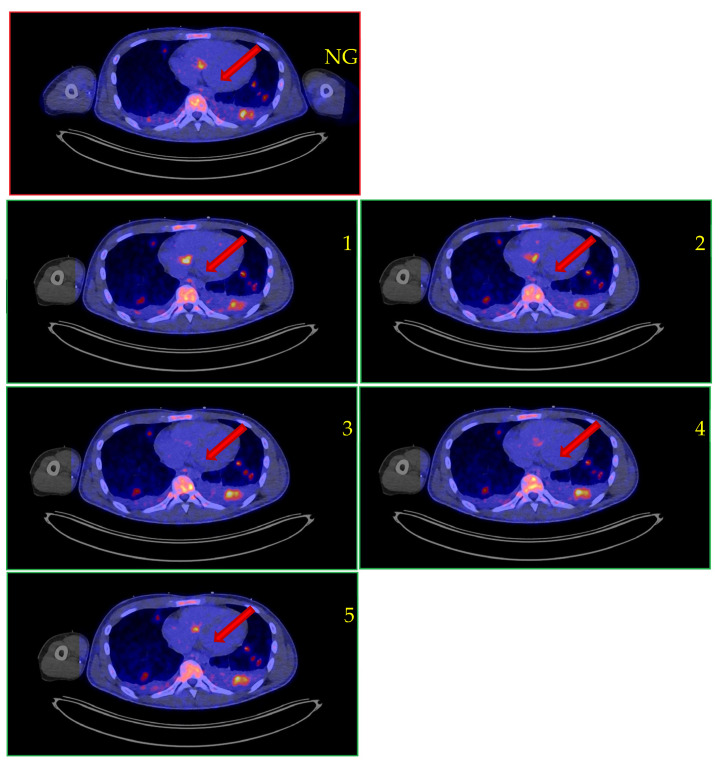
[^18^F]FDG PET/CT suspected native valve endocarditis of the tricuspid valve, as shown by the non-gated sequence (top image) and with conventional cardiac gating. Legend: NG, non-gated [^18^F]FDG PET/CT; 1–5: gated images. Arrows, tricuspid valve vegetation. Note the movement of the vegetation on the gated image, confirming the association of the lesion with the tricuspid valve, and the multiple bilateral lung lesions from septic embolization.

**Figure 3 diagnostics-13-03146-f003:**
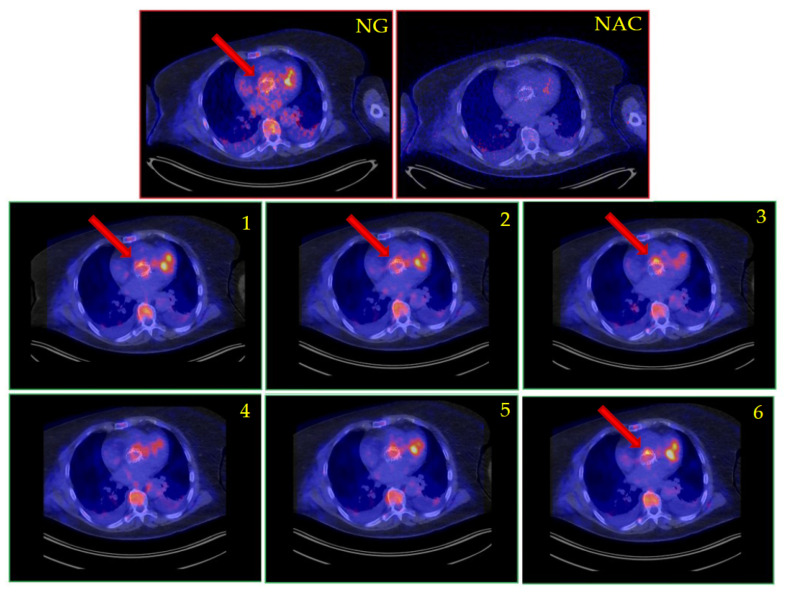
[^18^F]FDG PET/CT suspected prosthetic valve endocarditis of a TAVI valve, as shown by the non-gated sequence (top image) and with conventional cardiac gating. Legend: NG, non-gated [^18^F]FDG PET/CT; NAC, non-attenuation-corrected ungated image; 1–6: gated images. Arrows, valve lesion. Note the disappearance of valvular uptake on NAC images, the difference in cardiac background and the distinct movement patterns of ventricular activity and valve activity on the gated images, increasing the suspicion of infection.

**Table 1 diagnostics-13-03146-t001:** Patient characteristics.

Patient Characteristics (*n* = 7)	
Age, Years (median, IQR)	76 (22–80)
Gender Male	5 (71%)
BMI (Median, IQR)	23.8 (19.4–31.3)
Diabetes mellitus, *n* (%)	1 (14%)
Prosthetic valves	4 (57%)
Cardiac-implanted devices	1 (14%)
Use of BioGlue^®^	0 (0%)
HFLC diet	7 (100%)
Heparin prior to scan	5 (71%)
Myocardial suppression Good Moderate Poor	5 (71%) 2 (29%) 0 (0%)
Blood glucose level at PET/CT <11 mmol/L (198 mg/dl) ≥11 mmol/L	7 (100%) 0 (0%)
Leucocytes < 4 or >10 × 10^9^/L	3 (43%)
CRP ≥ 5 mg/L	6 (86%)
eGFR (median, IQR)	56 (52–124)
Duration IV antibiotics (mean, SD) days	5.4 (2.9)
Positive blood cultures	6 (86%)

IQR: interquartile range; BMI: body mass index; HFLC: high-fat, low-carbohydrate; CRP: C-reactive protein; eGFR: estimated glomerular filtration rate; IV: intravenous; SD: standard deviation.

**Table 2 diagnostics-13-03146-t002:** Overview of patient findings.

Cultures of removed material (surgery)	Not applicable	Not applicable	Not applicable	*Cutibacterium avidum* ^3^	Not applicable	Not applicable	Not applicable
Blood cultures	*Staphylococcus saprophyticus*	*Staphylococcus aureus*	*Staphylococcus aureus*	Negative	*Enterococcus faecalis*	*Staphylococcus aureus*	*Aggregatibacter aphrophilus*
Interval implantation and PET/CT	Valve and CIED: 11 months	Not applicable	Not applicable	Valve: 10 years	Valve: 4 years	CIED: 13 years	Valve: 10 years
CIED	Yes	No	No	No	No	Yes	No
**Valve type & location**	Prosthetic (Aortic)	Native (Aortic)	Native (Tricuspid)	Prosthetic (Aortic)	Prosthetic (Aortic)	Native (All valves)	Prosthetic (Pulmonic)
**Age (Y)**	77	76	18	78	74	23	22
**Gender**	Male	Female	Male	Male	Female	Female	Male
**Patient No.**	* **1** *	* **2** *	* **3** *	* **4** *	* **5** *	* **6** *	* **7** *
**Final Dx** **IE**	Yes	No	Yes	Yes	Yes	No	Yes
**Image quality**	Non-gated: 3 Gated: n.a. CardioFr: 4	Non-gated: 4 Gated: 4 CardioFr: 4	Non-gated: 3 Gated: 4 CardioFr: n.a.	Non-gated: 2 Gated: 2 CardioFr: n.a.	Non-gated: 2 Gated: n.a. CardioFr: 3	Non-gated: 4 Gated: 4 CardioFr: 4	Non-gated: 3 Gated: 4 CardioFr: 4
**PET results**	Non-gated: IE confirmed Gated: Not available CardioFreeze^TM^: IE confirmed (fl+)	Non-gated: No IE Gated: No IE CardioFreeze^TM^: No IE	Non-gated: IE confirmed Gated: IE confirmed (fl+) CardioFreeze^TM^: Not available	Non-gated: Equivocal Gated: Equivocal CardioFreeze^TM^: Not available	Non-gated: No IE Gated: Not available CardioFreeze^TM^: IE confirmed	Non-gated: No IE Gated: Not available CardioFreeze^TM^: No IE	Non-gated: IE confirmed Gated: IE confirmed (fl+) CardioFreeze^TM^: IE confirmed (fl+)
**TTE, TEE, CTA**	TTE: IE confirmed TEE: IE confirmed CTA: IE confirmed	TTE: No IE ^1^ TEE: Not performed ^2^ CTA: No IE	TTE: IE confirmed TEE: IE confirmed CTA: Not performed	TTE: No IE TEE: Not performed CTA: No IE	TTE: No IE TEE: IE confirmed CTA: Not performed	TTE: No IE TEE: Not performed CTA: No IE	TTE: No IE TEE: Not performed ^4^ CTA: Not performed
**Patient No.**	** *1* **	** *2* **	* **3** *	** *4* **	** *5* **	** *6* **	** *7* **

Legend: Image quality score 0–4 (uninterpretable, poor, moderate, good, excellent), (fl+) fluttering lesion observed consistent with a vegetation; ^1^ TTE image quality was poor in this patient; ^2^ patient could not undergo TEE due to neck complaints caused by spondylodiscitis; ^3^
*Cutibacterium avidum*: formerly *Propionibacterium avidum* (presence established using 16S-rDNA PCR with sequencing); ^4^ TEE was not performed because the suspected valve was a pulmonic (Contegra^TM^) valve, which is better visualized using TTE.

## Data Availability

Study data are available through the corresponding author upon reasonable request.

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
