# Peer review of "Improved [18F]FDG PET/CT Diagnostic Accuracy for Infective Endocarditis Using Conventional Cardiac Gating or Combined Cardiac and Respiratory Motion Correction (CardioFreezeTM)"

_diagnostics, 2023, doi:10.3390/diagnostics13193146_

Round 1

Reviewer 1 Report

In summary, your paper is well-structured and informative, addressing an important clinical issue. To enhance the manuscript, consider making the title more specific, adding quantitative results to the abstract, introducing motion correction techniques earlier in the introduction, organizing the Materials and Methods section into subsections, specifying statistical tests used, and presenting key quantitative results in a summary or table in the Results section. These improvements will help readers quickly grasp the paper's main contributions. Overall, it's a promising study with valuable clinical implications.

Abstract:

The abstract is well-structured and concise, effectively summarizing the paper's objectives, methods, key findings, and implications. However, it lacks specific quantitative results. Including some numerical findings or statistical outcomes in the abstract could provide readers with a clearer sense of the study's significance.

Introduction:

The introduction provides a thorough background on infective endocarditis and its diagnostic challenges. It successfully establishes the context and importance of the study. However, consider briefly introducing motion correction techniques and their relevance to the topic earlier in the introduction. This would help readers understand why motion correction is being investigated in this context.

Materials and Methods:

The Materials and Methods section is well-structured and comprehensive. It includes essential details about patient selection, ethical approval, imaging protocols, and diagnostic criteria. To enhance clarity, consider breaking down this section into subsections for patient selection, imaging protocol, image analysis, and diagnostic criteria. Additionally, specify the statistical tests used in the analysis to provide a clear understanding of the data analysis process.

Results:

The Results section effectively presents the study's findings, providing detailed descriptions of each patient's case and supporting images. The use of figures and images enhances the clarity of the results. However, it would be beneficial to provide a summary or a table that presents the main quantitative results, such as sensitivity, specificity, or other relevant metrics, for easier reference. This would help readers quickly grasp the study's primary outcomes.

Discussion:

The Discussion section offers a comprehensive analysis of the study's findings and their implications. It effectively compares conventional single-gate gating with CardioFreezeTM and highlights the advantages of motion correction in improving image quality and diagnostic accuracy. The mention of patient burden and radiation is relevant. Continue to acknowledge the study's limitations, as you have, and emphasize the need for larger prospective cohorts to validate the findings.

Conclusions:

The Conclusions section provides a succinct summary of the key findings and their significance. It rightly underscores the potential of motion correction techniques in diagnosing infective endocarditis and calls for further research. 

Here are some suggestions:

1. Statistical Analysis:

Consider performing statistical tests to assess the significance of the differences observed in the study. For example, you could use paired t-tests or Wilcoxon signed-rank tests to compare the image quality scores between gated and non-gated images.

Calculate and report sensitivity, specificity, positive predictive value, negative predictive value, and diagnostic accuracy for both the conventional gating and CardioFreezeTM techniques. This will provide quantitative measures of their diagnostic performance.

If applicable, use statistical tests to compare the radiation exposure between the two techniques. This can help quantify the potential benefits in terms of reduced radiation exposure to patients.

Conduct a power analysis to determine if the sample size in your study is adequate to detect significant differences. If not, discuss the limitations related to sample size and suggest a target sample size for future studies.

2. Additional Experiments:

Consider conducting a comparative analysis of the cost-effectiveness of the two motion correction techniques. This could include an evaluation of the additional time, resources, and equipment required for CardioFreezeTM compared to conventional gating.

If feasible, perform a subgroup analysis to investigate whether certain patient characteristics (e.g., age, comorbidities, type of endocarditis) influence the effectiveness of motion correction. This could provide insights into patient selection for these techniques.

Explore the impact of motion correction on interobserver variability. Involve multiple readers in the image analysis and assess the agreement between their interpretations with and without motion correction.

If you have access to historical data, consider a retrospective analysis to compare outcomes and diagnostic accuracy before and after the implementation of motion correction techniques in your clinical practice.

3. Statistical Software:

Utilize statistical software such as SPSS or R for data analysis. These tools provide a wide range of statistical tests and visualization options to enhance your analysis and presentation.

By incorporating these additional analyses and experiments, you can strengthen the scientific rigor of your study, provide more robust evidence for the benefits of motion correction in diagnosing infective endocarditis, and contribute to a more comprehensive understanding of the topic. Be sure to clearly report the methods and results of any new analyses or experiments in your manuscript.

By addressing these comments and incorporating the suggested improvements, your paper can become more comprehensive and robust, contributing significantly to the field of diagnosing infective endocarditis with [18F] FDG PET/CT motion correction techniques.

The quality of English language in the manuscript is generally good. The text is well-structured, with a clear flow of ideas. However, there are a few areas where minor improvements can be made:

In the abstract, the sentence "This may improve the sensitivity of PET/CT" seems to have a typographical error. It should be revised to "This may improve the sensitivity of PET/CT."

In the introduction, the sentence "This difficulty is further compounded by the rapid, prolonged, and intensive treatment the disease requires to improve patient outcomes" could be rewritten for clarity, such as: "This difficulty is exacerbated by the need for prolonged and intensive treatment to improve patient outcomes."

In the materials and methods section, the phrase "confirmed the diagnosis of endocarditis, which was missed on non-gated PET" could be rephrased for greater clarity, like: "confirmed the diagnosis of endocarditis, which had been missed in the non-gated PET."

In the discussion section, the sentence "Cardiac gating and/or CardioFreezeTM achieved improved interpretability and confirmed an otherwise missed diagnosis of IE in this case series" could be made clearer by rephrasing it as: "Cardiac gating and/or CardioFreezeTM improved interpretability and confirmed a diagnosis of IE that had otherwise been missed in this case series."

Overall, the manuscript is well-written and communicates the research effectively, but these minor adjustments can enhance clarity.

Author Response

Dear reviewer,

We want to thank you for the time and effort spent in evaluating our manuscript. Likewise we thank you for the constructive remarks and suggestions and we have implemented these throughout the manuscript. A point-by-point response follows below (bold for comments, blue in Italics for responses):

General comments

In summary, your paper is well-structured and informative, addressing an important clinical issue. To enhance the manuscript, consider making the title more specific, adding quantitative results to the abstract, introducing motion correction techniques earlier in the introduction, organizing the Materials and Methods section into subsections, specifying statistical tests used, and presenting key quantitative results in a summary or table in the Results section. These improvements will help readers quickly grasp the paper's main contributions. Overall, it's a promising study with valuable clinical implications.

We thank you for your kind words. As advised, we incorporated more quantitative results into both the abstract and the main body of the manuscript. Furthermore, the Results section has been organized in subsections and a summary of the quantitative results is added at the end of the Results section.

Abstract:

The abstract is well-structured and concise, effectively summarizing the paper's objectives, methods, key findings, and implications. However, it lacks specific quantitative results. Including some numerical findings or statistical outcomes in the abstract could provide readers with a clearer sense of the study's significance.

Thank you for the suggestion: we incorporated specific quantitative results in the abstract section.

Introduction:

The introduction provides a thorough background on infective endocarditis and its diagnostic challenges. It successfully establishes the context and importance of the study. However, consider briefly introducing motion correction techniques and their relevance to the topic earlier in the introduction. This would help readers understand why motion correction is being investigated in this context.

Thank you for the remark. The first mention of cardiac motion and its correction has been moved to  the beginning of the  introduction for improved flow of this section.

Materials and Methods:

The Materials and Methods section is well-structured and comprehensive. It includes essential details about patient selection, ethical approval, imaging protocols, and diagnostic criteria. To enhance clarity, consider breaking down this section into subsections for patient selection, imaging protocol, image analysis, and diagnostic criteria. Additionally, specify the statistical tests used in the analysis to provide a clear understanding of the data analysis process.

Thank you for the suggestion: as stated above, subsections have been added, along with the statistical tests used

Results:

The Results section effectively presents the study's findings, providing detailed descriptions of each patient's case and supporting images. The use of figures and images enhances the clarity of the results. However, it would be beneficial to provide a summary or a table that presents the main quantitative results, such as sensitivity, specificity, or other relevant metrics, for easier reference. This would help readers quickly grasp the study's primary outcomes.

Thank you for your comment. We agree that the final part of section Results would benefit from reporting quantitative results, which we have done as much as applicable. The manuscript now includes a statistical comparison of image quality with and without cardiac motion correction. Due to the heterogeneity of this series with both suspected NVE and PVE, results regarding FDG PET/CT’s overall performance were reported descriptively.

Discussion:

The Discussion section offers a comprehensive analysis of the study's findings and their implications. It effectively compares conventional single-gate gating with CardioFreezeTM and highlights the advantages of motion correction in improving image quality and diagnostic accuracy. The mention of patient burden and radiation is relevant. Continue to acknowledge the study's limitations, as you have, and emphasize the need for larger prospective cohorts to validate the findings.

Thank you for the kind words and the suggestions. This is incorporated in the manuscript, and the limitation section has been expanded to emphasize the need of future studies.

Conclusions:

The Conclusions section provides a succinct summary of the key findings and their significance. It rightly underscores the potential of motion correction techniques in diagnosing infective endocarditis and calls for further research. 

Thank you for the kind words.

Here are some suggestions:

  1. Statistical Analysis:

Consider performing statistical tests to assess the significance of the differences observed in the study. For example, you could use paired t-tests or Wilcoxon signed-rank tests to compare the image quality scores between gated and non-gated images.

Calculate and report sensitivity, specificity, positive predictive value, negative predictive value, and diagnostic accuracy for both the conventional gating and CardioFreezeTM techniques. This will provide quantitative measures of their diagnostic performance.

If applicable, use statistical tests to compare the radiation exposure between the two techniques. This can help quantify the potential benefits in terms of reduced radiation exposure to patients.

Conduct a power analysis to determine if the sample size in your study is adequate to detect significant differences. If not, discuss the limitations related to sample size and suggest a target sample size for future studies.

We thank the reviewer for this useful suggestion, and have added Wilcoxon’s signed rank test to compare the image quality scores. This study is intended as an exploratory study, which precluded further statistical analysis due to heterogeneity and sample size.  Regarding the overall PET/CT performance we have not made direct comparisons due to the risk of confounding as mentioned previously. In essence, for the cases in which both conventional gating and CardioFreeze were performed, the resulting images were indistinguishable from one another, so comparisons between the two techniques were not performed afterwards. Radiation burden was also identical for the two gating strategies, as both only needed one extra low-dose CT sequence  for attenuation correction.

We have expanded the Discussion with regard to power analysis and sample size calculations as follows: “Due to the exploratory nature of the study, no power analyses were performed. An important consideration for future studies is that the prevalence of IE at the time of [18F]FDG-PET/CT is not known, due to differences in timing of PET/CT and other data (e.g. NVE vs. PVE, pathogen, other diagnostic tests performed, prior antimicrobial treatment, etc.). These factors are, among others, determinants of the a priori chance of disease presence and of [18F]FDG-PET/CT diagnostic performance. Although we estimated a sample size of 58 per arm based on our preliminary data using Yates’ Continuity correction in a paired study design (β=80%, α=0.05), this remains a rough estimate that would require interim analysis for more accurate calculation.”

  1. Additional Experiments:

Consider conducting a comparative analysis of the cost-effectiveness of the two motion correction techniques. This could include an evaluation of the additional time, resources, and equipment required for CardioFreezeTM compared to conventional gating.

If feasible, perform a subgroup analysis to investigate whether certain patient characteristics (e.g., age, comorbidities, type of endocarditis) influence the effectiveness of motion correction. This could provide insights into patient selection for these techniques.

Explore the impact of motion correction on interobserver variability. Involve multiple readers in the image analysis and assess the agreement between their interpretations with and without motion correction.

If you have access to historical data, consider a retrospective analysis to compare outcomes and diagnostic accuracy before and after the implementation of motion correction techniques in your clinical practice.

Thank you for the suggestions. We agree these are interesting avenues for future research and while we currently do not have the necessary data to conduct these additional experiments, we hope they will be part of future studies. Subgroup analyses will require significant patient group numbers, but considering the very low patient burden associated with cardiac motion correction, this should be feasible at a later stage.

  1. Statistical Software:

Utilize statistical software such as SPSS or R for data analysis. These tools provide a wide range of statistical tests and visualization options to enhance your analysis and presentation.

By incorporating these additional analyses and experiments, you can strengthen the scientific rigor of your study, provide more robust evidence for the benefits of motion correction in diagnosing infective endocarditis, and contribute to a more comprehensive understanding of the topic. Be sure to clearly report the methods and results of any new analyses or experiments in your manuscript.

By addressing these comments and incorporating the suggested improvements, your paper can become more comprehensive and robust, contributing significantly to the field of diagnosing infective endocarditis with [18F] FDG PET/CT motion correction techniques.

Thank you for the suggestions. As shown in the manuscript, we have used SPSS to conduct the additional statistical analyses.

Comments on the Quality of English Language

The quality of English language in the manuscript is generally good. The text is well-structured, with a clear flow of ideas. However, there are a few areas where minor improvements can be made:

In the abstract, the sentence "This may improve the sensitivity of PET/CT" seems to have a typographical error. It should be revised to "This may improve the sensitivity of PET/CT."

In the introduction, the sentence "This difficulty is further compounded by the rapid, prolonged, and intensive treatment the disease requires to improve patient outcomes" could be rewritten for clarity, such as: "This difficulty is exacerbated by the need for prolonged and intensive treatment to improve patient outcomes."

In the materials and methods section, the phrase "confirmed the diagnosis of endocarditis, which was missed on non-gated PET" could be rephrased for greater clarity, like: "confirmed the diagnosis of endocarditis, which had been missed in the non-gated PET."

In the discussion section, the sentence "Cardiac gating and/or CardioFreezeTM achieved improved interpretability and confirmed an otherwise missed diagnosis of IE in this case series" could be made clearer by rephrasing it as: "Cardiac gating and/or CardioFreezeTM improved interpretability and confirmed a diagnosis of IE that had otherwise been missed in this case series."

Overall, the manuscript is well-written and communicates the research effectively, but these minor adjustments can enhance clarity.

Thank you for your kind words and the textual suggestions. They have all been incorporated in the manuscript.

Reviewer 2 Report

An interesting and informative manuscript that has clinical merit.  However, there are editing issues that the authors should consider and address.  The following are suggestions/comments regarding those issues.  Line 28, "This may improve the ...".  Line 44, Define CIED.  Line 48, "and use of this technique as part ...".  Lines 59 & 60, "... implanted electronic devices related to endocarditis ...".  Line 89, "... prior to PET/CT, intervals between the ...".  Line 105, "... CardioFreeze TM, in which the sequences were performed ...".  Line 141, "An overview of the patient characteristics ...".  Line 154, "... confirmed IE with intense ...".  Line 200, "Two years prior to the current episode, the ...".  Line 214, "... confirmed by TEE."  Line 217, "... of endocarditis due to the absence of ...".  Line 231, "... seen on TEE and both ...".  Line 277, "... sequence could also work, which may ...".  Line 293, "... of vegetation fluttering.  CardioFreeze TM performed ...".

The manuscript is well-written.

Author Response

Dear reviewer,

On behalf of the review group I want to thank you for the time and effort spent in evaluating our manuscript. Likewise we thank you for the constructive suggestions and we have implemented these throughout the manuscript.

An interesting and informative manuscript that has clinical merit.  However, there are editing issues that the authors should consider and address.  The following are suggestions/comments regarding those issues.  Line 28, "This may improve the ...".  Line 44, Define CIED.  Line 48, "and use of this technique as part ...".  Lines 59 & 60, "... implanted electronic devices related to endocarditis ...".  Line 89, "... prior to PET/CT, intervals between the ...".  Line 105, "... CardioFreeze TM, in which the sequences were performed ...".  Line 141, "An overview of the patient characteristics ...".  Line 154, "... confirmed IE with intense ...".  Line 200, "Two years prior to the current episode, the ...".  Line 214, "... confirmed by TEE."  Line 217, "... of endocarditis due to the absence of ...".  Line 231, "... seen on TEE and both ...".  Line 277, "... sequence could also work, which may ...".  Line 293, "... of vegetation fluttering.  CardioFreeze TM performed ...".

Thank you for these helpful suggestions. They have all been incorporated into the manuscript.

Comments on the Quality of English Language

The manuscript is well-written.

Thank you for your kind words.

Round 2

Reviewer 1 Report

I have reviewed the revised version of your manuscript, and I’m pleased to inform you that all of my previous concerns and comments have been satisfactorily addressed. The paper has significantly improved, and it is now in an acceptable form for publication. Thank you for your hard work and responsiveness during the revision process.